# LANGUAGE-DRIVEN 3D HUMAN POSE ESTIMATION: GROUNDING MOTION FROM TEXT DESCRIPTIONS

## ABSTRACT

In an NBA game scenario, consider the challenge of locating and analyzing the 3D poses of players performing a user-specified action, such as attempting a shot. Traditional 3D human pose estimation (3DHPE) methods often fall short in such complex, multi-person scenes due to their lack of semantic integration and reliance on isolated pose data. To address these limitations, we introduce Language-Driven 3D Human Pose Estimation (L3DHPE), a novel approach that extends 3DHPE to general multi-person contexts by incorporating detailed language descriptions. We present Panoptic-L3D, the first dataset designed for L3DHPE, featuring 3,838 linguistic annotations for 1,476 individuals across 588 videos, with 6,035 masks and 91k frame-level 3D skeleton annotations. Additionally, we propose Cascaded Pose Perception (CPP), a benchmarking method that simultaneously performs language-driven mask segmentation and 3D pose estimation within a unified model. CPP first learns 2D pose information, utilizes a body fusion module to aid in mask segmentation, and employs a mask fusion module to mitigate mask noise before outputting 3D poses. Our extensive evaluation of CPP and existing benchmarks on the Panoptic-L3D dataset demonstrates the necessity of this novel task and dataset for advancing 3DHPE. Our dataset can be accessed at `https://languagedriven3dposeestimation.github.io/`.

## 1 INTRODUCTION

3D human pose estimation (3DHPE) Pavlakos et al. (2017); Pavllo et al. (2019); Sun et al. (2017); Zheng et al. (2021); Sun et al. (2022); Su et al. (2022) seeks to accurately localize joints and reconstruct the body's representation within a 3D coordinate system based on input images or videos. This capability to derive detailed motion and geometric information about the human body underpins a plethora of applications, including video understanding Wang et al. (2021), sports analytics Bridgeman et al. (2019), and human-robot interaction Xu et al. (2020). However, traditional approaches have primarily concentrated on identifying poses without integrating high-level semantic knowledge, which is crucial for meaningful human communication. Despite recent efforts Zhang et al. (2020); Mazzia et al. (2022); Chi et al. (2022); Delmas et al. (2024); Lin et al. (2024); Feng et al. (2024) to link human pose with semantic information, these methods still face significant limitations. The actions they consider are typically simplistic, which results in ordinary and limited semantic descriptions. Moreover, the input videos are usually restricted to scenarios involving a single individual engaged in sports activities, thereby limiting the practical applicability of 3DHPE. In reality, human motion is characterized by diverse and detailed descriptions, highlighting an emerging need for 3D human pose estimation to address more complex, real-world, and multi-person scenarios. This necessitates a shift towards incorporating motion grounding, a concept that connects dynamic human movements with rich semantic context, enabling a more comprehensive understanding of human activities in natural and multifaceted environments.

In response to these challenges, we introduce a novel task: *Language-Driven 3D Human Pose Estimation* (L3DHPE), which can also be referred to as *3D Motion Grounding*. It extends the original 3DHPE problem to encompass more general multi-person scenes and explores the semantic interplay between human poses and language expressions. L3DHPE addresses a more demanding yet practical problem, aiming to reconstruct 3D pose sequences for individuals based on detailed language descriptions that capture aspects such as appearance, behavior, and body movements.

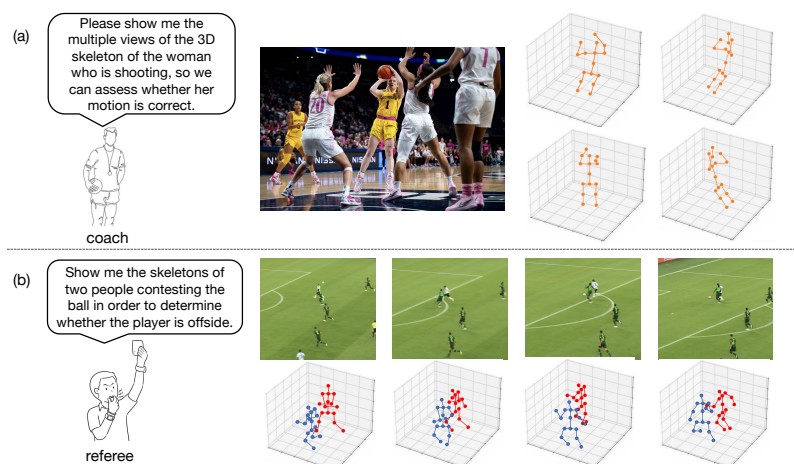

Figure 1: Example applications of the L3DHPE task: Sports coaching and officiating.

We present two notable real-world examples of L3DHPE applications. In the first example (Fig. 1 (a)), a coach can focus on specific players and view them from multiple angles using language descriptions. By integrating language understanding with 3D pose reconstruction, L3DHPE enables real-time analysis of player movements, significantly enhancing feedback and performance. The second example (Fig. 1 (b)) involves sports officiating. A referee can use language input to quickly isolate the relevant players' skeletons, simplifying decisions like offside calls by filtering out irrelevant players.

Given the scarcity of pose estimation datasets with precise and richly contextual linguistic annotations, we developed Panoptic-L3D, the first L3DHPE dataset, to propel advancements in this field. Panoptic-L3D includes 588 videos featuring 1,476 targeted individuals, 3,838 language descriptions, 6035 mask images, and corresponding 91k frame-level 3D skeleton annotations for all individuals. Each video is meticulously selected to ensure the presence of at least two individuals, with clearly visible upper bodies and significant movements by at least one individual. Annotators provide detailed, long-form descriptions of individuals based on their appearance, behavior, and body movements, ensuring specificity to eliminate ambiguity. In addition to 3D skeleton annotations for pose information, we provide individual object masks using the pretrained SAM model Kirillov et al. (2023) for coarse-level pose guidance. Fig. 2 showcases examples from our Panoptic-L3D dataset, showing its comprehensive and detailed annotations.

Building upon the proposed Panoptic-L3D dataset, a naive solution for our task is to apply referring video object segmentation methods to videos. This approach takes both language and video information as input to generate masks of the described objects, which are then used in single-person 3D human pose estimation methods. However, this method has several significant drawbacks: Firstly, referring video object segmentation methods do not leverage pose information, leading to a poor understanding of poses and incorrect segmentation of individuals, thus reducing the accuracy of 3D human pose estimation. Secondly, in multi-person interactions, occlusions between individuals often occur, and masks may be incomplete, introducing noise that is amplified during 3D pose estimation, further degrading accuracy. Finally, the workflows of both the reference video object segmentation and single-person 3D human pose estimation methods are complex. Simply combining them results in an excessively cumbersome processing flow, limiting their practicality in real-life applications.

To address these issues, we propose Cascaded Pose Perception (CPP), a benchmarking method for the task we introduce. CPP jointly performs language-driven mask segmentation and 3D pose output within a single model. Initially, the model is trained to learn 2D pose information. We then design a body fusion module that utilizes pose information to assist the network in segmenting masks for specified individuals. Unlike traditional 3DHPE methods, we do not rely solely on the predicted human root nodes to output 3D poses. Instead, a mask fusion module is incorporated to help the model identify individuals from human masks while minimizing the impact of mask noise.

Our contributions are summarized below:

- We introduce Language-Driven 3D Human Pose Estimation, a novel yet practical 3D human pose estimation setting that incorporates language guidance. This setting emphasizes the text-motion interaction, which holds great potential in several real-life applications.

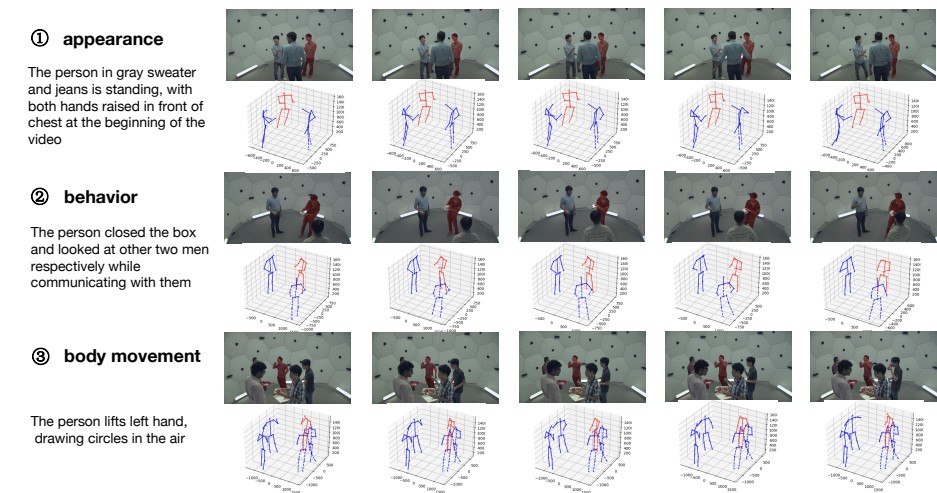

Figure 2: Examples of videos, descriptions, masks, and 3D skeleton of our Panoptic-L3D dataset.

- We construct Panoptic-L3D, the first dataset for L3DHPE, containing 3,838 linguistic annotations for 1,476 individuals across 588 videos, with 6,035 masks and 91k frame-level 3D skeleton annotations. We also propose a baseline, CPP, providing potential solutions for L3DHPE task.
- We extensively evaluate CPP and four RVOS benchmarks along with 3DHPE benchmarks on Panoptic-L3D. Experimental results demonstrate the drawbacks of previous methods and underscore the necessity of the L3DHPE task and the Panoptic-L3D dataset.

## 2 RELATED WORKS

**3D Human Pose Estimation Datasets.** We summarize various public 3D Human Pose Estimation datasets in Tab. 1. HumanEva-I Sigal et al. (2010) contains 7 calibrated video sequences synchronized with 3D body poses obtained from a motion capture system. Human3.6M Ionescu et al. (2013) includes 3.6 million human poses with corresponding frames, offering precise 3D human joint positions and high-resolution videos captured at 50 Hz by a motion capture system. CMU Panoptic Joo et al. (2015) features extensive social interactions with up to eight characters per video, making it a significant choice for our annotations. 3DPW (3D Poses in the Wild) Von Marcard et al. (2018) is the first dataset with accurate 3D poses for evaluation in the wild, using the Video Inertial Poser (VIP) method to combine images and IMU readings. EgoBody Zhang et al. (2022) and EgoPW Wang et al. (2022) focus on egocentric human pose estimation from a first-person perspective. As discussed in Sec. 3.2, these datasets focus on 3D human gestures without mask assistance or language guidance. Our Panoptic-L3D dataset provides expressive linguistic descriptions and segmentation masks, promoting close interactions between dynamic human poses and rich semantic context.

**Monocular 3D Human Pose Estimation Methods.** Monocular 3D Human Pose Estimation (3DHPE) aims to reconstruct the 3D coordinates of individuals from a single camera view. One challenging aspect is depth estimation from a single view. Generally, multiple camera views are required to infer the exact depth of individuals. However, the size of individuals and camera parameters can be used to approximate their depth, as explored in studies like Lee & Kim (2019); Li et al. (2019); Li & Snavely (2018). There are two approaches to monocular 3D pose estimation: single-stage and two-stage approaches. Single-stage methods Pavlakos et al. (2017); Mehta et al. (2017); Sun et al. (2017); Kanazawa et al. (2018); Sun et al. (2018) directly localize 3D human joints from input images or videos. Two-stage methods Akhter & Black (2015); Park et al. (2016); Moreno-Noguer (2017); Yang et al. (2018); Su et al. (2022); Park et al. (2023) first estimate 2D poses or utilize pre-trained 2D pose estimators, then lift them to 3D space. Since estimating 2D poses is easier than 3D poses, models can benefit from the reliable results of 2D pose estimation. Due to its good performance, we also utilize a two-stage approach in our model. However, traditional methods lack the guidance of language, which our approach incorporates to enhance 3D pose estimation and interaction.

**Referring Video Segmentation.** Referring Video Object Segmentation (RVOS) aims to segment target objects described by given text throughout an entire video clip. A2D Gavrilyuk et al. (2018) first

Figure 3: The pipeline for dataset collection and annotation involves several steps. First, we obtain candidate videos from the CMU Panoptic dataset. These videos are then cut and filtered into informative clips, which are used for subsequent annotation. We implement a multi-round check and re-annotation process to ensure the accuracy and validity of both the text descriptions and masks.

proposed the RVOS task. Advanced RVOS methods Liu et al. (2021); McIntosh et al. (2020); Wang et al. (2020); Botach et al. (2022) explore various pipelines to effectively aggregate and align visual, temporal, and linguistic information in videos and text. URVOS Seo et al. (2020) introduced a large-scale RVOS benchmark utilizing attention mechanisms and mask propagation. Liang et al. Liang et al. (2021) presented a top-down approach, first detecting all target trajectories and then selecting target objects through matching between language and trajectory features. Recently, MTTR Botach et al. (2022) and ReferFormer Wu et al. (2022) utilized query-based end-to-end frameworks to decode objects from multi-modal features, achieving outstanding performance. However, these methods segment prominently appearing objects without focusing on humans, lacking incorporation of human body pose information. Our approach combines human body pose information, leading to superior performance in understanding human actions.

## 3 PANOPTIC-L3D DATASET

Our objective is to enable diverse and fine-grained 3D human pose estimation for in-the-wild scenes. To achieve this, we introduce Panoptic-L3D, the first publicly available benchmark designed for language-driven 3D human pose estimation. Panoptic-L3D incorporates multi-modal information, including videos, language descriptions, and segmentation masks. The annotation pipeline of the Panoptic-L3D dataset is summarized in Fig. 3.

### 3.1 DATASET CONSTRUCTION

**Video and 3D Skeleton Collection.** To construct the language-driven 3D human pose estimation dataset, we utilize videos from the Panoptic Joo et al. (2015) dataset, a large-scale collection of indoor human videos that provides 1.5 million 3D skeleton annotations for individuals engaged in social activities. To simulate complex multi-person interactions found in real life, we select fourteen activity entries that feature more than one person and span five social interaction categories: Haggling, Ultimatum, Toddler, Musical Instruments, and Special Events. For each activity entry, we obtain two variants from different camera viewpoints (cameras No.16 and No.30) to ensure the robustness of 3D human pose estimation.

Subsequently, these videos are separated into clips and filtered according to the following rules:

- **R1**: The length of the video clips is standardized to ensure an average length of 5 seconds. This duration allows individuals to perform a series of actions without being overly complex, making it suitable for describing the behavior of characters using sentences.
- **R2**: To ensure the accuracy of language descriptions referring to the target individuals, each video clip must include at least two participants with fully exposed upper bodies, and at least one individual must be undergoing significant movements. This requirement focuses on perceptible limb movements, providing necessary motion information that matches the descriptions in 3D pose recognition and estimation.

Fig. 4 shows some invalid and valid samples. By rigorously separating and filtering the videos based on the above rules, we obtained 588 candidate videos with their corresponding 3D skeletons from the original Panoptic dataset, with an average duration of 5.17 seconds.

**Annotation and Validation for Linguistic Expression.** We employ over 10 annotators to label individuals in the videos according to our developed linguistic expression annotation system and guidelines. Specifically, each annotator is required to provide three sentences describing the attributes (i.e., appearance, behavior, and body movements) of each person visible from the waist up.

To ensure concise language descriptions, we establish the following guidelines for annotators:

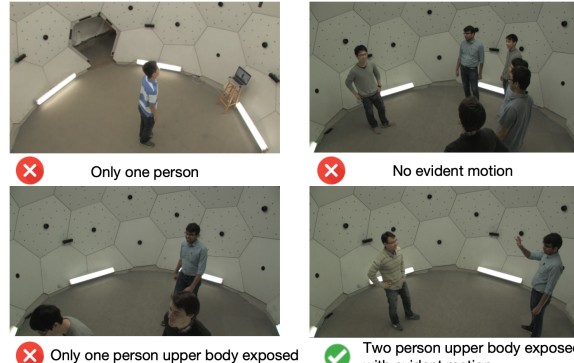

Figure 4: Examples of invalid and valid data.

- **Shared Rules:** 1) Only individuals visible from the waist up should be considered valid subjects for description. 2) Each description must uniquely refer to only one person in the given video. 3) Unobtrusive attributes should be omitted to avoid ambiguity.
- **Rules for Describing Appearance:** Annotators describe the appearance based on the first frame of the video, focusing on four specified elements: clothes, posture, gender, and initial pose. To simulate variability in individual perception, two of these four elements are uniformly sampled for each individual, and annotators are required to base their descriptions on the selected attribute pair. This procedure enhances the diversity and informativeness of appearance descriptions.
- **Rules for Describing Behaviors & Body Movements:** Linguistic expressions can describe behaviors across multiple frames, encompassing both fleeting and prolonged actions. Annotators must describe an individual's behavior and body movements after watching the entire video, ensuring comprehensive coverage. Descriptions of behaviors must exclude appearance information, while descriptions of body movements should focus solely on the motion itself, excluding interactions with environmental elements (e.g., cups, food, pens).

For validation, we follow an interactive game-like approach inspired by ReferIt Kazemzadeh et al. (2014), involving two participants in an alternating validation process. Initially, the original video and its corresponding descriptions are presented to the first validator, who must identify the target person referred to by the statements. If the chosen target matches the annotator's target, the annotated video is retained. If there is a discrepancy, the sample is forwarded to a second validator for further review. If the second validator finds any ambiguity, the description is revised. If ambiguity persists, the sample is discarded to maintain the accuracy of descriptions.

By adhering to these validation criteria, we ensure the unique referential integrity and high quality of language statements in our dataset, thereby enabling more robust evaluations and comparisons of different methods.

**Annotation and Validation for Individual Masks.** The mask annotations for each referred individual in our video data are generated using the advanced 2D segmentation model SAM Kirillov et al. (2023). We utilize the projected joint points of the upper body (i.e., neck and head top) from the 3D skeleton provided by the original Panoptic dataset as coarse point prompts for SAM. For each video, we generate a coarse mask every 15 consecutive frames.

However, some misprojected 3D joints may occur during mask generation due to overlapping between individuals. Additionally, SAM may focus only on the upper part of the referred person since the joints are mostly located in the upper body. To validate and correct the mask annotations, we perform a multi-round check and re-annotation process, ensuring completeness and consistency between the masked individual and the corresponding description.

**Dataset Split.** The dataset is divided into training, validation, and test sets, consisting of 3,005, 312, and 521 sentences, respectively, along with their corresponding videos, masks, and 3D skeletons.

## 3.2 DATASET ANALYSIS

**Comparison with Existing Datasets.** The Panoptic-L3D dataset consists of 588 videos, 1,476 target individuals, 3,838 linguistic descriptions, 6,035 mask images, and 91k per-frame 3D skeletons of all

Table 1: Statistics of representative 3d human pose estimation datasets. Our proposed dataset Panoptic-L3D has larger number of person in one video. More importantly, Panoptic-L3D has language and masks annotations while previous 3d human pose estimation datasets don't include. Panoptic-L3D enables the investigation of language-guided 3d pose estimation.

| Dataset | No. of frames/videos | No. of person | Language | Masks |
|---------|---------------------|---------------|----------|-------|
| HumanEva-I Sigal et al. (2010) | 12 sequences | 1 | ✗ | ✗ |
| Human3.6M Ionescu et al. (2013) | 3.6M frames | 1 | ✗ | ✗ |
| MARCOnI Elhayek et al. (2016) | 12 sequences | 1 or 2 | ✗ | ✗ |
| CMU Panoptic Joo et al. (2015) | 1.5M frames | 1 to 8 | ✗ | ✗ |
| Total Capture Trumble et al. (2017) | 1.892M frames | 1 | ✗ | ✗ |
| 3DPW Von Marcard et al. (2018) | 60 sequences | 1 or 2 | ✗ | ✗ |
| EgoBody Zhang et al. (2022) | 219k frames | 1 or 2 | ✗ | ✗ |
| EgoPW Wang et al. (2022) | 318k frames | 1 or 2 | ✗ | ✗ |
| **Our Panoptic-L3D** | 588 sequences / 91k frames | 2 to 7 | ✓ | ✓ |

individuals. Statistical comparisons are presented in Tab. 1. Compared to existing 3D human pose estimation datasets, our Panoptic-L3D dataset offers several notable advantages.

Firstly, Panoptic-L3D provides expressive linguistic descriptions and segmentation masks for individuals, which markedly differ from previous datasets that only offer 3D skeletons and action categories. The inclusion of linguistic descriptions is crucial for facilitating language-guided 3D pose estimation. Additionally, the videos in the Panoptic-L3D dataset are more practical and provide a better simulation of real-life scenarios.

As shown in Fig. 5, the video samples feature more characters and complex interactions among multiple individuals. The carefully clipped and filtered videos also ensure better alignment with the provided linguistic expressions, benefiting the comprehensive evaluation under our proposed language-guided 3D pose estimation setting.

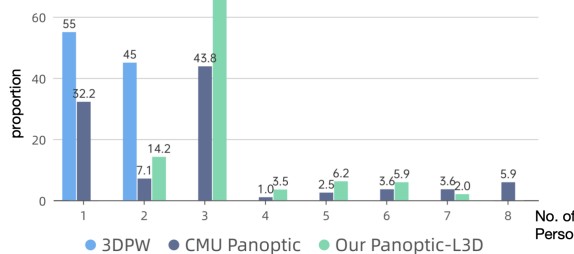

Figure 5: Distribution of the number of people present in videos for Panoptic Joo et al. (2015), 3DPW Von Marcard et al. (2018), and our dataset.

**Linguistic Expression.** We introduce a novel set of linguistic expressions that describe human motion in terms of appearance, behavior, and body movements, facilitating multi-modal interaction in 3D human pose estimation tasks. The linguistic descriptions are categorized into three distinct statements, each focusing on one of the three aspects, enabling systematic evaluation of the model's comprehension of varied language descriptions.

Fig. 6(a) shows the distribution of these categories, highlighting the balanced representation of diverse data types. While behavioral descriptions encompass a wide range of activities, body movement descriptions specifically address joint movements, imposing significant demands on model accuracy. The lengths of these linguistic expressions are depicted in Fig. 6(b), ranging from 6 to 39 words, demonstrating the diversity of linguistic expressions and presenting additional challenges for models.

We utilize word clouds to visualize the differences between descriptions in the behavioral and body movement aspects. As shown in Fig. 6(c) and 6(d), behavioral descriptions emphasize semantic actions such as "put", "talk", and "listen", while body movement descriptions focus on specific joints and movements, e.g., "right", "hands", and "arms".

## 4 CASCADED POSE PERCEPTION

**Problem Definition.** Given a video $V = \{v_i\}_{i=1}^L$, the corresponding camera parameters $P$, and the referring text description $T$, the goal of L3DHPE is to estimate the 3D skeleton $S \in \mathbb{R}^{15 \times 3}$ for the individual referred to by $T$.

**Overview.** The most relevant tasks are 3D human pose estimation (3DHPE) and referential video object segmentation (RVOS). Recent progresses in 3DHPE focus on accurately determining the spatial

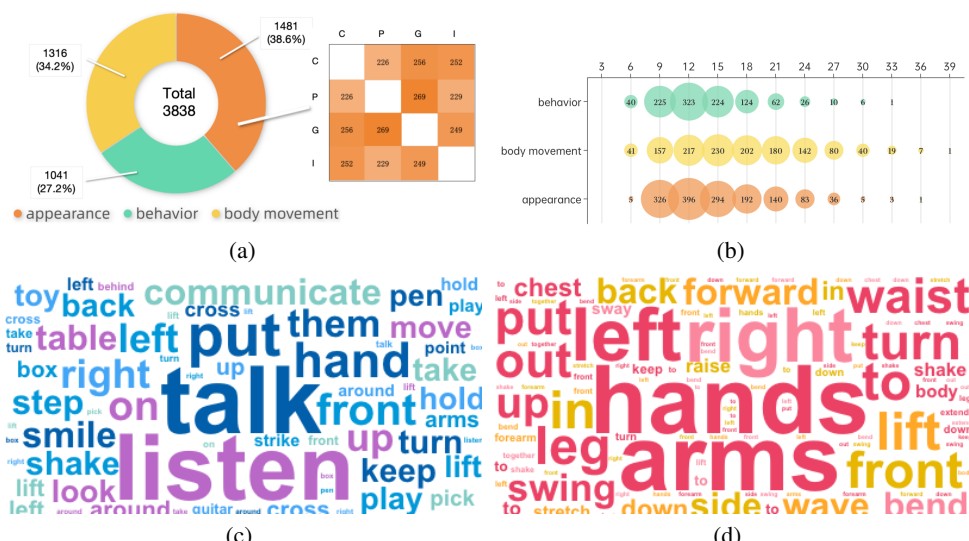

Figure 6: Statistics of annotations: (a) The number of description annotations and their distribution across various categories, i.e., **C**lothes, **P**osture, **G**ender, and **I**nitial pose. (b) Length distribution of descriptions, highlighting the diversity of Panoptic-L3D. Word clouds of the words in the Panoptic-L3D dataset are presented in (c) for behavioral descriptions and (d) for body movement descriptions.

positions of joints in 3D space, while less effort has been made to explore multi-modal interactions incorporating linguistic information. Conversely, RVOS models primarily emphasize the integration of language and appearance-level information but often overlook behavior-level information, leading to a diminished capacity to capture text-motion interaction for accurate 3D pose estimation.

To address these limitations, we propose a baseline, Cascaded Pose Perception (CPP), for language-driven 3D human pose estimation. CPP integrates pose features with video features in a two-stage manner. It first learns the 2D pose information and generates multiple levels of information, including root depth maps, 2D joints maps, and box detection maps, to support 3D human pose estimation. We then design a body fusion module to utilize this pose information to generate masks for specified individuals. Based on the generated masks, we further incorporate a 3D root estimator and a mask fusion block to improve the precision of 3D human pose estimation and reduce the impact of mask noise. The overall architecture of CPP is illustrated in Fig. 7.

**Method.** Initially, following Voxelpose Tu et al. (2020), we employ a backbone of 2D pose estimation to extract 2D pose information from the relevant frame of the input video. This procedure yields box detection maps $M_b \in \mathbb{R}^{W \times H \times 4}$, root (waist) depth maps $M_r \in \mathbb{R}^{W \times H}$, and 2D joints maps $M_j \in \mathbb{R}^{W \times H \times N_j}$ for all individuals in the frame. Here, $N_j$ indicates the number of human joints, and $W, H$ are the width and height of the output maps.

Next, we extract both video and textual features and design a *body fusion block* to integrate them with the 2D pose information, specifically the 2D joints maps $M_j$. The body fusion block blends the video features with pose information, enhancing the understanding of pose and the precision of individual segmentation.

The integrated features are then processed by a transformer-based auto-encoder to capture comprehensive information about all individuals present. Specifically, the human box detection maps $M_b$ are utilized to create object queries $O^q \in \mathbb{R}^{N_q \times D}$ that encapsulate each person's information, where $N_q$ is the number of object queries and $D$ is the dimension of object queries. By providing precise locational data, our method efficiently eliminates ambiguity in masks, achieving more precise identification and extraction of specified human features. These object queries are further transformed into bounding boxes, masks, and categories for all individuals, unified for outputting the candidate masks and corresponding confidence scores. The mask with the highest confidence is then selected as the final mask $M \in \mathbb{R}^{W \times H}$ for the referred person, which is also a side output of our CPP.

After deriving the referring mask $M$, the human box detection maps $M_b$, root depth maps $M_r$, and camera parameters $P$ are transmitted to the 3D root estimator, which determines the 3D positions of

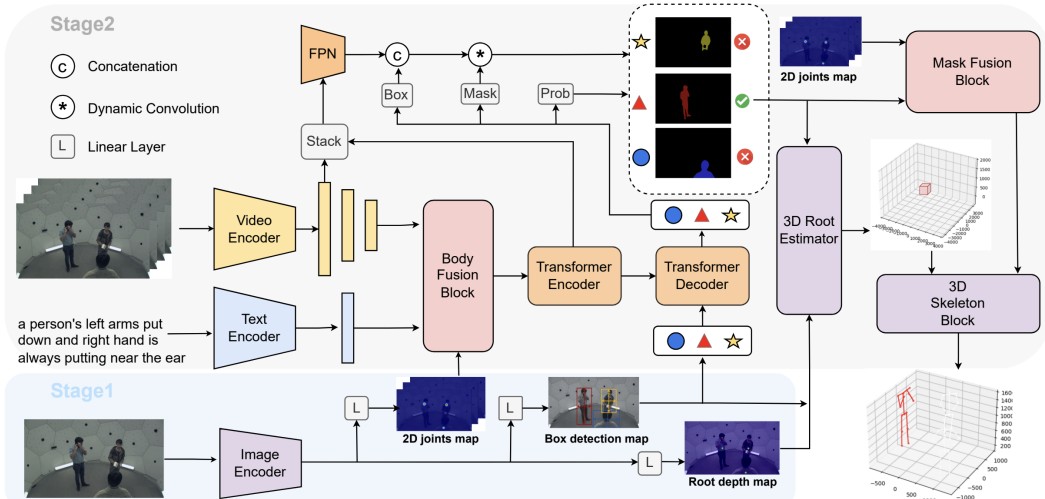

Figure 7: The overview of the proposed baseline approach, Cascaded Pose Perception (CPP). Given a video and corresponding textual description, an image encoder first generates spatial intermediate representations (e.g., 2D joints, bounding boxes, and depth maps). These features are fused with text features using a Body Fusion Block. The fused features and bounding boxes generate object queries, which are transformed into masks and confidence scores with the video features. Finally, a Mask Fusion Block and a 3D Root Estimator predict skeleton features and root position, which are processed by the 3D Skeleton Block for the final 3D skeleton estimation.

the human root nodes. Instead of relying solely on the 3D human root nodes for 3D pose estimation, we incorporate a *mask fusion block* for integrating the referring masks and human joint maps for all individuals, providing additional spatial features for understanding the human pose. The use of masks for identifying the referred person and selecting the individual's joints enables the model to pinpoint the person's nodes with enhanced accuracy and clarity. Finally, a 3D skeleton block is adopted to estimate a complete 3D human skeleton $S \in \mathbb{R}^{15 \times 3}$.

## 5 EXPERIMENTS

**Implementation Details.** The architectures of text encoder, video encoder, and pose encoder in our framework are pretrained RoBERTa Liu et al. (2019), Tiny Swin-Transformer Liu et al. (2022), and ResNet-152 He et al. (2016), respectively. The number of object queries is set to 10 by default. During training and testing, we feed the model windows of $w = 6$ frames. The models are trained using a 24GB NVIDIA 4090 GPU. Both stage 1 and stage 2 are trained for 10 epochs with a learning rate of 1e-4; each epoch for stage 1 and stage 2 takes approximately 1.5 hours and 2 hours, respectively. The input frames are resized to have a minimum size of 512 pixels on the shorter side and a maximum length of 960 pixels on the longer side to ensure efficient memory usage on the GPU.

**Evaluation Metrics.** The metrics employed in our proposed L3DHPE task are twofold: the Percentage of Individuals Correctly Identified (PICI) and the Mean Per Joint Position Error (MPJPE-L) in millimeters. PICI calculates the proportion of correct cases in referring to individual identification. MPJPE-L measures the accuracy of 3D skeleton estimation by calculating the mean Euclidean distance between the predicted and ground truth joint positions across all joints. The calculation approach for MPJPE-L is the same as that for MPJPE (usually used in 3DHPE task). The added 'L' indicates that this evaluation protocol simultaneously considers the tasks of text-referred person identification and minimizing the predicted pose's deviation. Note that when the identified person is not the actual person referred to in the text description, the predicted joints exhibit significantly larger variations compared to the ground truth, leading to a larger value of MPJPE-L than the normal MPJPE. We believe MPJPE-L is a more straightforward and comprehensive metric for evaluating methods in our L3DHPE task. A detailed discussion of these metrics is provided in the supplementary.

**Quantitative Results.** We conduct a quantitative comparison between CPP and existing four RVOS+3DHPE baselines for language-driven 3D human pose estimation on the Panoptic-L3D

Table 2: Quantitative evaluations for L3DHPE on the Panoptic-L3D dataset. The best results are highlighted in **bold**. App., Beh. and Mov. are the abbreviation of appearance, behavior and body movement, respectively, denoting three kinds of attributes of our language annotations.

| Methods | Pub. | PICI | | | MPJPE-L | | |
|---|---|---|---|---|---|---|---|
| | | App. | Beh. | Mov. | App. | Beh. | Mov. |
| URVOS Seo et al. (2020)+VP | ECCV20 | 35.5% | 35.1% | 34.9% | 1298.0 | 1323.2 | 1325.9 |
| MTTR Botach et al. (2022)+VP | CVPR22 | 36.1% | 35.7% | 35.6% | 1223.5 | 1256.7 | 1261.3 |
| Referformer Wu et al. (2022)+VP | CVPR22 | 37.0% | 36.6% | 36.5% | 1104.6 | 1133.6 | 1139.4 |
| SOC Luo et al. (2024)+VP | NeurIPS24 | 38.7% | 37.6% | 37.5% | 965.1 | 992.8 | 1004.3 |
| CPP (Ours) | - | **42.6%** | **41.3%** | **41.2%** | **853.7** | **885.4** | **886.3** |

dataset. Since RVOS methods only produce segmented masks for the referred object, we equip them with the state-of-the-art 3D human pose estimation (3DHPE) method Virtualpose Su et al. (2022) (VP) to estimate 3D skeletons. As shown in Tab. 2, our proposed CPP captures more interaction between the given text description and the pose of the referring individuals. Naively combining RVOS models with a 3D human pose estimation backbone does not produce satisfactory performance in the language-driven 3D human pose estimation task, demonstrating the necessity and potential contribution of our Panoptic-L3D dataset in promoting the development of a more coherent and language-perceptible 3D pose analysis system.

To investigate the advantages of selecting the referred subject first, rather than performing 3DHPE for all subjects, we implemented a variant of our method called 3DHPE+Refer. This approach performed 3DHPE for all subjects and then selected the referred in-

Table 3: Quantitative evaluations for 3DHPE+Refer and CPP on the Panoptic-L3D dataset. The best results are highlighted in **bold**.

| methods | PICI | | | MPJPE-L | | | FPS |
|---|---|---|---|---|---|---|---|
| | App. | Beh. | Mov. | App. | Beh. | Mov. | |
| 3DHPE+Refer | 37.1% | 37.6% | 40.1% | 1091.4 | 1003.9 | 901.3 | 2.0 |
| CPP (Ours) | **42.6%** | **41.3%** | **41.2%** | **853.7** | **885.4** | **886.3** | **3.5** |

dividual. Specifically, we removed the text branch and estimated the 3D skeletons of every individual in each video frame. Next, we applied the Hungarian Algorithm to associate individuals across multiple frames. After detecting all individuals, we stacked several transformer blocks to fuse the text information with the skeleton features. The fused features were then used to generate probabilities for each individual, ultimately selecting the subject with the highest probability. The results are shown in Tab. 3. Our pipeline outperforms the 3DHPE+Refer approach in our L3DHPE task, because estimating poses for all individuals before grounding introduces cumulative pose estimation errors. Additionally, 3DHPE+Refer requires explicit aggregation of individuals across the entire video, which introduces another potential source of error in subject detection. In contrast, CPP maintains better temporal continuity with the video backbone, reducing this risk. Furthermore, CPP is more efficient than 3DHPE+Refer, as it avoids the time-consuming process of estimating poses for all individuals in the video.

We also evaluate the performance of CPP compared to existing state-of-the-art methods on the traditional 3DHPE task. Since there are no text descriptions provided, we design a variant of CPP by discarding the text encoder. We adopt Virtualpose Su et al. (2022) as the backbone 3D estimation network and equip it with additional mask interaction. Previous state-of-the-art models are trained on multiple datasets simultaneously.

Table 4: Comparison of state-of-the-art methods on the CMU Panoptic dataset for the 3D human pose estimation task, using MPJPE (mm) as the metric. The best and second-best results are highlighted in **bold** and underline, respectively.

| Method | Haggling | Mafia | Ultimatum | Pizza | Mean |
|---|---|---|---|---|---|
| MubyNet Zanfir et al. (2018) | 72.4 | 78.8 | 66.8 | 94.3 | 78.1 |
| SMAP Zhen et al. (2020) | 63.1 | 60.3 | 56.6 | 67.1 | 61.8 |
| BEV Sun et al. (2022) | 90.7 | 103.7 | 113.1 | 125.2 | 108.2 |
| VirtualPose Su et al. (2022) | 54.1 | 61.6 | 54.6 | 65.4 | 58.9 |
| POTR-3D Park et al. (2023) | 60.0 | **57.0** | 55.5 | **58.9** | **57.8** |
| CPP (Ours) | **54.0** | 60.4 | **54.6** | 62.1 | **57.8** |

For a fair comparison, we employ a selective parameter-tuning strategy to fine-tune our model, training only the blocks associated with mask processing. Specifically, we first fine-tune Detectron2 Wu et al. (2019) to generate human masks. Next, we use and freeze the parameters of the Image Encoder and 3D Skeleton Block from Virtualpose. Finally, we train the 3D Root Estimation and Mask Fusion Block to estimate the 3D skeletons. As presented in Tab. 4, CPP achieves competitive performance with specifically designed 3DHPE models, demonstrating the effectiveness of incorporating spatial information in 3D pose understanding.

Table 5: Ablation study of CPP.

| ID | Text | Body Fus. | Mask Fus. | PICI | | | MPJPE-L | | |
|---|---|---|---|---|---|---|---|---|---|
| | | | | App. | Beh. | Mov. | App. | Beh. | Mov. |
| i | ✓ | ✗ | ✗ | 40.3% | 39.4% | 39.2% | 918.1 | 936.7 | 937.8 |
| ii | ✓ | ✓ | ✗ | 42.5% | 41.2% | 41.2% | 887.3 | 920.4 | 922.3 |
| iii | ✓ | ✓ | ✓ | 42.6% | 41.3% | 41.2% | 853.7 | 885.4 | 886.3 |

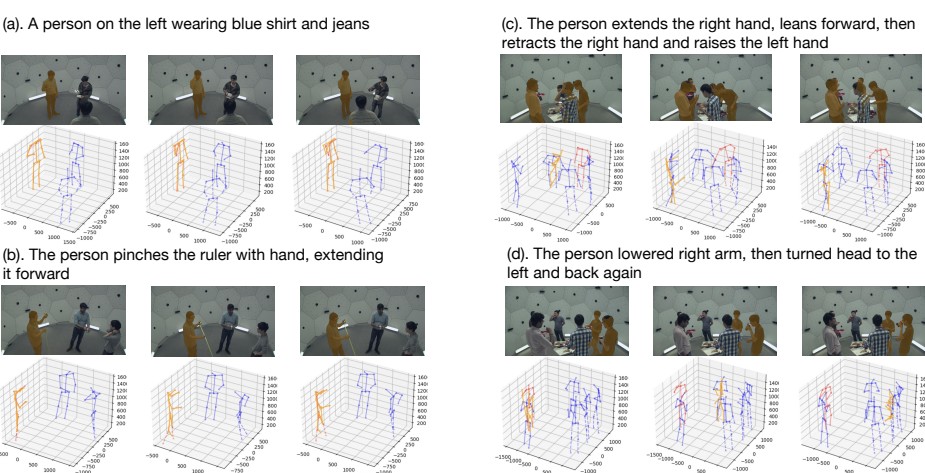

Figure 8: Visualization of successful (left) and failed (right) cases of CPP. The output mask and 3D skeleton of CPP are shown in orange, while the ground truth 3D skeletons are shown in red.

**Ablation Study of CPP.** In Tab. 5, we present an ablation study for the proposed CPP. Specifically, we design two variants of CPP to evaluate the functions of the feature interaction in the Body Fusion Block and the Mask Fusion Block. Compared to the vanilla baseline, which uses only the input language queries for 3D human pose estimation, equipping the Body Fusion Block with additional pose information from the 2D joint features enables more accurate spatial localization for the referred individual, thereby improving the PICI by 2% (i vs. ii). Furthermore, utilizing the Mask Fusion Block introduces spatial locational features from the referring mask, facilitating more precise 3D human pose estimation with an average improvement of 34.7 in MPJPE-L (ii vs. iii).

**Visualizations.** We provide visualizations of both successful and failed cases of CPP. As shown in Fig. 8 (a) and (b), CPP effectively generates clear boundary masks and accurate 3D poses based on the given expressions. However, CPP may fail to detect the referred individual in complex scenarios with multiple individuals, substantial occlusion, and swift body movements. The results in Fig. 8 (c) and (d) demonstrate the challenging nature of Panoptic-L3D, emphasizing the importance of capturing the semantic interaction between poses and language expressions under severe occlusion.

## 6 CONCLUSION AND DISCUSSION

Imagine a future where machines understand human actions and intentions through language. This paper introduces Language-Driven 3D Human Pose Estimation and the Panoptic-L3D dataset, blending video data with linguistic annotations. The Cascaded Pose Perception baseline demonstrates the power of integrating textual descriptions with visual data. While acknowledging the manual nature of our annotation process as a challenge for future work, we see L3DHPE enhancing fields from sports analytics to advanced robotics. By releasing Panoptic-L3D and CPP, we invite the research community to build on our foundation, exploring and improving language-guided 3D pose estimation.

**Limitations and Future Work.** There are many interesting research directions and remaining challenges to be addressed with the Panoptic-L3D dataset and L3DHPE task. These include but are not limited to: (i) designing more efficient models to increase running speed, (ii) designing more accurate and elegant models to recognize and understand complex motions, (iii). designing robust cross-modal fusion methods to better leverage motion information and language information.

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
