# OpenReview forum: "Language-driven 3D Human Pose Estimation: Grounding Motion from Text Descriptions"
_ICLR.cc/2025/Conference — Submitted to ICLR 2025_

### Official Review · Reviewer_H6EN · 2024-10-18

**Soundness:** 3
**Presentation:** 4
**Contribution:** 3
**Rating:** 3
**Confidence:** 5

**Summary:**

This paper introduces Language-Driven 3D Human Pose Estimation, which estimates the Human pose with the text description.

This paper also labels the text on the CMU Panoptic dataset to support their research.

**Strengths:**

This paper proposed a novel fusion model to handle the different types of data, which are from the images and texts. This paper also gives a strong and clear process of how they process and label the data, which supports their research.

**Weaknesses:**

1. the application of Language 3D human pose estimation

In this paper, the authors show some cases where we need 3D pose estimation with the guide of language. But, in most of cases, current LLM can directly handle the problem to give the semantic segmentation for pose estimation unless the paper's model can have much higher performance than top-down multi-person pose estimation. The performance problem I will talk in detail later. Besides, In this paper, the only function of language is just to help humans focus on or select the person they want in the scene. That's week to show the language function in pose estimation.

2. Evaluation Metrics
This paper emphasizes that MPJPE-L is the better metric for language and pose together, but going through the supplement doc, I still did not see the specific formula of MPJPE-L. If just a weighted sum with PICI and MPJPE, it's unfair to set up a weight to compare with other SOTA models.

3. The backbone
This paper chooses the VP as the backbone of the network which is an old network for pose estimation. And in the experiment section, the performance is pretty lower compared with 2024 SOTA models. Thus, I think the author should consider at least 2024 SOTA models as the backbone and compared with the latest model. Besides, notably,  this paper compared only two separate SOTA models, such as Referformer Wuetal.(2022) + VP. I can understand why chose two separate models, but I think you still need to compare the fusion model for pose estimation which is more fair to check your performance.

**Questions:**

1. what's the formula of MPJPE-L? Please use the mathematic way to prove why it's better to use it instead of MPJPE.

2. In this first train stage why did not choose the 2D GT or CPN's results as the input following other 3D pose estimation, so that we can check the performance fairly. Besides, even if you chose VP as the 2D predictor, why did you get the 3D output from VP?

3. Because you have much more requirements of data for training, training data in this paper is much smaller than others, how do you ensure the training results can avoid the overfit? Do you try other datasets to test your model to prove your model is not overfit?

4. you use lots of pre-trained models. what's the total size of the whole model? Do you compare it with other SOTA models?

---

### Official Review · Reviewer_B8AM · 2024-10-26

**Soundness:** 1
**Presentation:** 2
**Contribution:** 2
**Rating:** 3
**Confidence:** 4

**Summary:**

This paper introduces the Language-Driven 3D Human Pose Estimation task and a Panoptic-L3D dataset for estimating the 3D pose of a specified target. It also presents a benchmarking method, CPP, which demonstrates promising performance on the Panoptic-L3D dataset.

**Strengths:**

1. The introduced task generates results for a user-specified person, which may benefit related fields.
2. The proposed CPP method is reasonable for this task.

**Weaknesses:**

1. The writing can be improved.
2. The discussion of related works is incomplete, missing many state-of-the-art methods.
3. The dataset includes mask annotations generated by SAM; how is their accuracy ensured? In addition, it is unclear whether mask supervision is necessary during training, and no segmentation comparison is provided.
4. The paper asserts that the introduced task can benefit sports, but if I understand correctly, the dataset includes only indoor videos, which does not support this claim. Additionally, to avoid misleading, Figure 1 should display examples from the Panoptic-L3D dataset.
5. The dataset lacks complex scenarios, diverse sports actions, and varied backgrounds, limiting its ability to demonstrate the significance and effectiveness of this work.
6. Some details are missing, for example, what Refer method is used in Table 3?
7. The visualizations are inadequate, showing only a few image examples without video results, despite this being a video-based task. Furthermore, the examples lack complexity and cannot effectively support the claim.

**Questions:**

See Weaknesses

**Details Of Ethics Concerns:**

CMU Panoptic Studio dataset is shared only for research purposes, and this cannot be used for any commercial purposes. The dataset or its modified version cannot be redistributed without permission from dataset organizers. However, the introduced dataset is built upon the CMU Panoptic dataset.

---

### Official Review · Reviewer_3wiE · 2024-10-27

**Soundness:** 3
**Presentation:** 2
**Contribution:** 3
**Rating:** 5
**Confidence:** 3

**Summary:**

This work presents a novel task: Language-Driven 3D Human Pose Estimation, which extends the original 3D human pose estimation to encompass more general multi-person scenes and explores the semantic interplay between human poses and language expressions. To support this task, a new dataset named Panoptic-L3D has been introduced. In addition, the authors also propose a unified model, Cascaded Pose Perception (CPP), a benchmarking method that simultaneously performs language-driven mask segmentation and 3D pose estimation within a unified model. Extensive experiments show the effectiveness of the proposed method.

**Strengths:**

1. This paper proposes a new task—Language-Driven 3D Human Pose Estimation (L3DHPE), which combines natural language processing with 3D human pose estimation in an interdisciplinary task.
2. This paper provides a new perspective for understanding human movements in complex multi-person scenarios, significantly extending existing methods of 3D human pose estimation.
3. The proposed Cascaded Pose Perception (CPP) method demonstrates how to integrate language and visual information within a single model for more accurate 3D pose estimation. The introduction of this method offers new ideas and benchmarks for designing models in similar tasks in the future.

**Weaknesses:**

1. Important details of the Cascaded Pose Perception are missing.
While the overall methodology is presented, the methods section does not offer sufficient details about the proposed body fusion block and mask fusion block. It remains unclear how the body fusion block and mask fusion block are designed to achieve their specific functions. Even after examining the anonymized code, I still couldn't find the relevant information.
2. Insufficient ablation studies.
The ablation studies only evaluate the entire body fusion block and mask fusion block, providing no insight into their internal designs. Moreover, it remains unclear whether the chosen modules, excluding the body fusion block and mask fusion block, are the most optimal options. Given that the authors used ResNet-152 as the image encoder (lines 411-413), I am curious to know if substituting it with a more powerful human pose network like CPN [1] or HRNet [2] would lead to improvement.
3. Inference speed. (Minor weakness)
Although the authors mention designing a more efficient model in the future work section (line 538), I want to know whether the inference speed of the current model can truly support the real-time analysis as the authors claimed in lines 72-73.

Reference
[1] Cascaded Pyramid Network for Multi-Person Pose Estimation
[2] Deep high-resolution representation learning for human pose estimation

**Questions:**

Please address my concerns in the weaknesses section.

---

### Meta-Review · Area_Chair_p5pR · 2024-12-20

**Metareview:**

All reviewers have given negative feedback, and the authors have not provided any response. It is recommended to reject the manuscript.

**Additional Comments On Reviewer Discussion:**

All reviewers have given negative feedback, and the authors have not provided any response. It is recommended to reject the manuscript.

---

### Decision · Program_Chairs · 2025-01-22

Reject